# Decreased ALFF and Functional Connectivity of the Thalamus in Vestibular Migraine Patients

**DOI:** 10.3390/brainsci13020183

**Published:** 2023-01-22

**Authors:** Xia Zhe, Min Tang, Kai Ai, Xiaoyan Lei, Xiaoling Zhang, Chenwang Jin

**Affiliations:** 1Department of Radiology, The First Affiliated Hospital of Xi’an Jiaotong University, Xi’an 710061, China; 2Department of MRI, Shaanxi Provincial People’s Hospital, Xi’an 710068, China; 3Department of Clinical Science, Philips Healthcare, Xi’an 710000, China

**Keywords:** resting state, functional connectivity, thalamus, vestibular migraine, amplitude of low-frequency fluctuation

## Abstract

Background: The thalamus has been reported to be associated with pain modulation and processing. However, the functional changes that occur in the thalamus of vestibular migraine (VM) patients remain unknown. Methods: In total, 28 VM patients and 28 healthy controls who were matched for age and sex underwent resting-state functional magnetic resonance imaging. They also responded to standardized questionnaires aimed at assessing the clinical features associated with migraine and vertigo. Differences in the amplitude of low-frequency fluctuation (ALFF) were analyzed and brain regions with altered ALFF in the two groups were used for further analysis of whole-brain functional connectivity (FC). The relationship between clusters and clinical features was investigated by correlation analyses. Results: The ALFF in the thalamus was significantly decreased in the VM group versus the control group. In the VM group, the ALFF in the left thalamus negatively correlated with VM episode frequency. Furthermore, the left thalamus showed significantly weaker FC than both regions of the medial prefrontal cortex, both regions of the anterior cingulum cortex, the left superior/middle temporal gyrus, and the left temporal pole in the VM group. Conclusions: The thalamus plays an important role in VM patients and it is suggested that connectivity abnormalities of the thalamocortical region contribute to abnormal pain information processing and modulation, transmission, and multisensory integration in patients with VM.

## 1. Introduction 

Vestibular migraine (VM) is a complex disorder with a one-year prevalence of approximately 2.7% [1]. It is characterized by recurrent vertigo and migraine episodes [2]. The disorder poses a substantial burden on the healthcare system because of the severity of its symptoms [1,3]. VM is primarily diagnosed based on clinical history and the absence of peripheral vestibulocochlear dysfunction. However, the precise pathophysiology of VM remains to be fully elucidated. Neuroimaging techniques may provide some preliminary insights into the pathophysiology of VM and facilitate the development of better diagnostic and treatment options for these patients.

In recent years, many neuroimaging techniques have been employed to evaluate VM patients. Recently, positron emission tomography (PET) studies have shown the involvement of the vestibulo-thalamocortical pathway in these patients [4]. Compared with PET, MRI, which does not have nuclear radiation and has a high spatial resolution, has been used as an effective tool to study the structure and function of patients with VM. Several MRI studies in which voxel-based morphometry (VBM) was used reported brain structure alterations in various regions related to vestibular processing and pain, including the temporal lobe [5], frontal and occipital regions [6], frontal lobe [7], and parieto-insular vestibular cortex [8], in VM patients. Additionally, a task functional magnetic resonance imaging (fMRI) study by Russo et al. [9] has shown increased activation in the thalamus of VM patients. In another task fMRI study, Teggi et al. [10] reported activation of the brain areas that are associated with the integration of visual and vestibular cues (the inferior parietal lobule, paracentral lobule, head of the caudate nucleus, superior frontal gyrus, left superior temporal gyrus, left parahippocampal gyrus, and lingual gyrus) in VM patients. Previous task fMRI studies have demonstrated the brain function mechanism in patients with VM. However, task-based fMRI-generated outcomes might reflect some aspect of the primary pathophysiological mechanism of disease or be attributable to adaptive actions in task execution. Resting-state fMRI in the absence of task conditions is easier to implement and better able to capture intrinsic functional brain differences between patients and healthy controls [11]. In resting-state fMRI studies, evaluations of both the ALFF and FC are regarded as important approaches. The ALFF is an effective indicator of blood-oxygenation-level-dependent (BOLD) signal fluctuations, which can be used to evaluate the amplitude of each voxel from the viewpoint of energy and reflect neuronal spontaneous activity levels in the resting state [12,13,14]. The ALFF reaches the best balance between test–retest reliability and replicability [15]. FC indicates the temporal correlation of BOLD signal fluctuation between spatially remote brain regions, which in turn can indicate the functional interactions of spatially distributed brain regions [16,17]. These two methods have been widely applied to evaluate brain disorders, including migraine [18,19], VM [14,20], and persistent postural-perceptual dizziness [21,22]. Several studies have revealed functional abnormalities in the brain regions related to nociceptive, vestibular, and visual processing, such as the putamen and lingual gyrus [20], as well as the cerebellum [14] in VM patients. 

Although Russo and colleagues [9] and Shin and colleagues [4] confirmed thalamus functional abnormalities among VM patients in fMRI and PET studies, the thalamus plays a major role in nociception and vestibular integration and as a relay station between the central pain and vestibular systems. It also has been proven to have an important role in cortico-cortical communication via the transthalamic transfer of information between cortical areas that are involved in vestibular processing [23]. FC changes in the thalamus have been detected in migraine. Previous studies which used resting-state functional MRI have described FC alterations in migraine between the thalamus and pain modulating cortical areas, which suggests that the thalamus may play an important role in migraine. These altered thalamocortical connectivity patterns may contribute to multisensory integration abnormalities and pain modulation [24,25]. Some previous studies demonstrated that activation in thalamic nuclei can contribute to the modulation of trigeminovasclar and other spinal nociceptive inputs [26,27]. The altered FC of thalamus regions with other brain areas would affect the multisensory integration and processing of pain, and the relationship between VM and thalamocortical connectivity needs further investigation.

In this study, we conducted ALFF and FC analyses with resting-state fMRI to investigate both the regional spontaneous activities and FC in patients with VM and healthy controls. We first determined the areas with abnormal ALFF and then estimated their resting-state FC with each voxel throughout the whole brain. Based on brain activity abnormalities in the thalamus reported in previous studies [4,9,28], we hypothesized that: (1) alterations in spontaneous brain activity of VM patients during the resting state may occur in the thalamus; (2) brain regions within thalamus cortical connectivity would be vulnerable in VM patients; and (3) these alterations would be linked to clinical parameters in VM patients.

## 2. Materials and Methods

### 2.1. Patients

A total of 28 right-handed VM patients (4 with aura and 24 without aura) were enrolled from the Shaanxi Provincial People’s Hospital, China, from January 2016 to October 2021. The patients were diagnosed based on the International Classification of Headache Disorder 3rd edition criteria [29] by an experienced neurologist. Patients were excluded if they were previously diagnosed with other audiovestibular, psychiatric, neurologic, or systemic disorders. All patients underwent MRI and routine neurologic and neuro-otological examinations; both procedures were performed during symptom-free intervals on the same day. Videonystagmography (VNG) recordings showed no peripheral vestibular dysfunction. During direct interviews using a standardized questionnaire and questions, the neurologist also recorded the following information about the patients: sex, age, disease duration, episode frequency, and scores for the visual analog scale (VAS), Migraine Disability Assessment Scale (MIDAS), Dizziness Handicap Inventory (DHI), and Head Impact Test-6 (HIT-6). Six VM patients received medications to prevent migraine as well as nonsteroidal analgesics. Twenty-two patients were not given any medication regularly. With the purpose of avoiding any possible pharmacological interference with BOLD signal changes, patients with VM did not take medication for at least 3 days before the fMRI scan.

Matched for handedness, age, and sex, the 28 healthy controls were recruited in the community. Individuals were excluded if they had migraine; neurologic, mental, or systemic disorders; chronic pain; previous vestibular neuritis; ischemic or hemorrhagic stroke; secondary somatoform vertigo; Meniere’s disease; or severe head trauma. There was no white matter lesion or structural abnormality in any of the patients, as shown on T2-weighted or fluid-attenuated inversion recovery imaging. All patients were right-handed and did not abuse drugs. The Shaanxi Provincial People’s Hospital’s ethics committee approved this study. Written informed consent from the participants was given prior to study enrollment.

### 2.2. Imaging Data Acquisition 

A 3.0 T scanner (Philips Ingenia, Best, The Netherlands), which was equipped with a phased-array head coil (16 channels), was used to obtain all the images. A high-resolution 3D magnetization-prepared rapid-acquisition gradient echo T1-weighted sequence covering the whole brain (332 sagittal slices) was collected, with a repetition time (TR), flip angle (FA), inversion time (TI), echo time (TE), matrix, slice thickness, and field of view of 1900 ms, 9°, 900 ms, 2.26 ms, 256 × 256, 1.00 mm (with no interslice gap), and 220 × 220 mm, respectively. Gradient echo-planar imaging was used for resting-state functional BOLD images, with a repetition time, echo time, number of slices, slice thickness, slice gap, field of view, matrix, flip angle, and number of volumes of 2000 s, 30 ms, 34, 4 mm, 0 mm, 230 × 230 mm, 128 × 128, 90°, and 200, respectively. The patients were required to close their eyes all the time and stay awake and calm during the entire scan in the resting state. Thereafter, we asked the patients whether they were awake throughout the scan.

### 2.3. Image Processing

The Data Processing Assistant for the Resting-State fMRI Advanced Edition toolbox was used for the preprocessing and processing of all functional images. To ensure the stability of the BOLD signal, the first 10 time points were not considered. To correct for interslice delays within each volume, slice time was corrected. The exclusion criteria were >1.5° head rotation in any direction, >1.5 mm head translation in any direction, or >0.2 mm volume-level mean framewise displacement [30]. With a standard DARTEL template provided by SPM12, we spatially normalized images into the MNI space and resliced them into a voxel size of 3 × 3 × 3 mm^3^. A Gaussian kernel (6 mm, full width at half maximum) was used to spatially smooth the data. Finally, a typical temporal bandpass filter was employed to filter all the images to decrease low-frequency drift and physiological high-frequency respiratory and cardiac noise.

### 2.4. ALFF Analysis

Previous preprocessed results were used for the ALFF analysis. For a given voxel, we performed a fast Fourier transform so that the time sequences were transformed to the frequency series, and then calculated the power spectrum’s square root. The ALFF value was defined as the average square root. We subtracted the average ALFF value from each voxel’s ALFF value and then divided the result by the whole-brain ALFF map’s standard deviation; in this way, the standard ALFF value was obtained in order to reduce individual differences among the patients. These analyses were performed using the DPABI software.

### 2.5. Functional Connectivity Analysis

Per the ALFF results, the region of interest (ROI) was the region that was significantly different between the two participant groups. Subsequently, Pearson’s correlation coefficients between the time courses for each ROI and the signal time series for each voxel across the whole brain were calculated to generate each participant’s ROI-FC map. A Fisher’s r-to-z transformation was conducted for the said map to generate a z-score FC map for an improved normality of correlation coefficients [31]. The DPABI software was used for these analyses.

### 2.6. Statistical Analysis

#### 2.6.1. Demographic and Clinical Data

The differences in age and years of education between the two participant groups were estimated using a two-sample *t*-test. A *χ^2^*-test was used to detect intergroup sex differences. A *p* value of <0.05 was considered to indicate statistical significance and the SPSS software package (version 22.0) was used for statistical analyses. 

#### 2.6.2. ALFF Analysis

The two-sample *t*-tests were performed within the DPABI to compare the maps of significant differences with covariates of age and sex. Multiple comparison correction was performed using a Gaussian random field at a cluster *p*-value < 0.05 (voxel *p* < 0.001).The surviving clusters were reported. Individual mean ALFF values for the surviving VM patient clusters were extracted for partial correlations with clinical indices, including disease duration; attack frequency; and VAS, MIDAS, HIT-6, and DHI scores, with covariates of age and sex. Multiple comparisons were made using Bonferroni corrections (*p <* 0.05/[6 × *n*], where n and 6 are the numbers of ROIs and behavioral measurements, respectively).

#### 2.6.3. Seed-Based FC Analysis

The two-sample *t*-tests were performed within the DPABI to compare the z-score FC (zFC) maps of significant differences in FC between the two participant groups, using covariates of age and sex. A Gaussian random field at cluster *p* < 0.05 (voxel *p* < 0.001) was used for multiple-comparison correction. The surviving clusters were reported. Individual mean FC values for the surviving clusters of VM patients were extracted to determine their partial correlations with clinical indices, with covariates of age and sex. Multiple comparisons were made using Bonferroni corrections (*p* < 0.05/[6 × *n*], where n and 6 represent the numbers of ROIs and behavioral measurements, respectively).

## 3. Results

### 3.1. Demographic and Clinical Data

Intergroup differences in the following parameters were not significant: age (*p =* 0.53), sex (*p =* 1), and years of education (*p =* 0.14). The results of the intergroup comparison are presented in Table 1. The VM group reported a migraine burden (moderate to severe) with mean VAS, HIT-6, and MIDAS scores of 5.07 ± 2.73, 51.86 ± 19.36, and 47.46 ± 43.58, respectively. The VM group had moderate scores on the vertigo scale, with the mean DHI score being 47.71 ± 16.04. Patients with VM had other symptoms including nausea (42.85%), vomiting (35.71%), phonophobia (67.85%) and photophobia (75%). Of the 14 (50%) patients who had recorded spontaneous vertigo, 9 (32.14%) reported positional vertigo, 1 (3.57%) reported visually induced vertigo, and 4 (14.29%) reported head-motion-induced vertigo.

### 3.2. ALFF Results

In the VM group, only the ALFF values in the thalamus were bilaterally decreased, as compared with those in the control group, as shown in Figure 1 and Table 2.

### 3.3. Seed-Based FC Results

Two clusters showed bilaterally decreased ALFF values in the thalamus. Thus, we selected these two clusters as seeds for further FC analysis. The left thalamus showed significantly weaker FC with the bilateral medial prefrontal cortex (mPFC), the bilateral anterior cingulum cortex (ACC), the left superior/middle temporal gyrus, and the temporal pole in the VM group. Therefore, there was no significant difference between the right thalamus and other brain areas in the two groups. (Table 3, Figure 2).

### 3.4. Correlation of Clinical Parameters with ALFF and FC 

The ALFF values in the left thalamus significantly and negatively correlated with VM episode frequency in the VM group (*r* = −0.768; *p* = 0.000, Bonferroni correction with *p* < 0.05/12 = 0.004, Figure 3). Abnormal ALFF values were not found to be correlated with scores on VAS, MIDAS, and DHI and disease duration in the VM group. Furthermore, FC alterations were not found to be significantly correlated with clinical parameters in the VM group.

## 4. Discussion

In this study, we found changes in functional abnormalities in multiple brain regions during the vertigo-free period in VM patients. Furthermore, in these patients, the ALFF decreased bilaterally in the thalamus. Additionally, seed-to-whole-brain voxel analyses showed a large decrease in FC between the left thalamus and cortical regions, including those associated with pain modulation and processing and transmitting and multisensory integration.

The thalamus is usually thought of as a relay region where information was transmitted to the cerebral cortex and feedback information from the cerebral areas was received [32,33]. The bilaterally decreased ALFF in the thalamus in this study indicate decreased spontaneous neuronal activity and functional impairment. Furthermore, the decreased ALFF values in the left thalamus negatively correlated with the frequency of VM attacks. These results suggest that VM might result in a weakening of neuronal activity in the thalamus, which is associated with the transmission and processing of pain information. A previous fMRI study reported the activation of the thalamus in VM patients during caloric vestibular stimulation [9]. Further comparisons between groups revealed that VM patients showed significantly greater thalamic activation than that shown by healthy controls and the degree of activation was correlated positively with the migraine attack frequency. A PET study of two VM patients by Shin et al. [4] showed increased bilateral thalamic activation during VM attacks. The activation of this region was thought to be due to increased activation of the vestibulo-thalamocortical pathways. These findings indicate the role of the thalamus as a major center of sensory relays with the vestibular pathways and its association with multisensory processing and integration, including vestibular, visual, and somatosensory inputs [34,35]. The inconsistencies in these findings may be explained by differences in the techniques used (resting-state and task fMRI), as well as in the state of disease (during attack-free periods and attack periods). Structural abnormalities have also been reported previously. A recent VBM study by Messina et al. [6] found decreased thalamic volume in VM patients versus healthy controls. Notably, the thalamus has been shown to have a major role in the trigeminovascular pathway and relaying of sensory information to multiple cortical networks [36]. It has been reported that the basis of VM maybe activation of the trigeminovascular pathway [37]. The activation of the trigeminovascular system may convey nociceptive information from the meninges to the central brain areas and cortex [38]. Because the thalamus is involved in the pain and sensory processing networks of the trigeminovascular pathway, its dysfunction probably also affects information transmission and the processing of pain information. These findings suggest a crucial role of the thalamus in VM.

Regarding the resting-state FC analysis based on brain regions with altered ALFF, VM patients showed decreased FC between the left thalamus and many other brain areas, including the bilateral medial prefrontal cortex, bilateral ACC, left superior/middle temporal gyrus, and temporal pole. The thalamus plays a critical role in cortico-cortical communication through the transfer of information between cortical regions [23]. Thalamocortical involvement has also been demonstrated in PET studies in VM [4]. The findings of our present study display that dysfunction in FC between the thalamus and cortical regions may disrupt thalamo–cortical interaction, which may contribute to abnormal multimodal sensory processing and perception during VM attacks. 

The mPFC, which is important for pain processing, pertains to the frontal cortex. It has been demonstrated to be involved in the modulation of pain catastrophizing [39], reduction in pain-induced sympathetic activity, and decreased facial expression of pain [40]. It has also been reported to be associated with the neuropathology of migraine. Migraine exhibited alterations in the frontal regions involved in nociception [41,42]. Consistent with this finding, Zhe et al. [8] found a significantly reduced gray matter volume in the mPFC compared with that in healthy controls in a VBM study. Furthermore, VM patients showed significantly abnormal ALFF in the prefrontal cortex compared with the healthy controls [20]. In our study, VM patients showed significantly reduced FC between the left thalamus and bilateral mPFC. We speculated that the dysfunction associated with the reduced FC between the left thalamus and bilateral mPFC may disturb the modulation of the process to pain and evaluation, which could contribute to the pathology of VM.

The ACC, which is located in the medial aspect of the cerebral cortex, is involved in emotional and affective processing [38]. Pain can lead to highly significant brain structure and function changes, with the most affected brain region being the ACC [15,43,44,45]. Vogt, Zhuo, Guo, and their colleagues [46,47,48] suggested that pain and unpleasant pain-related emotion can be relieved by the inhibition of the functional activity of the ACC. Recent VBM-based analyses of VM patients by Obermann et al. [5] have shown a decrease in the gray matter volume in the ACC. We found reduced FC in both sides of the ACC and the left thalamus. Therefore, we speculated that reduced functional thalamo-ACC pain pathway could reflect the emotional modulation and processing of pain, which might be associated with disrupted pain processing and modulation as a result of repeated pain attacks. 

The temporal lobe is closely associated with multisensory integration. The studies by Coppola, Yu, Zhang, and their colleagues [49,50,51] have shown that patients with migraine have structural temporal lobe abnormalities. Obermann and colleagues [5] and Messina and colleagues [6] have found altered gray matter volumes in the temporal lobe of VM patients, which indicated that this brain region is majorly involved in pain regulation. We also demonstrated that multiple areas of the temporal lobe, including the superior/middle temporal gyrus and temporal pole, were altered in VM patients. Furthermore, the temporal pole is an associative multisensory area that processes visual, olfactory, and auditory information [52,53,54]. The superior temporal gyrus is considered to be involved in the perception of emotions [55]. The middle temporal gyrus seems to be relevant functionally for vestibular compensatory mechanisms and is strongly interconnected with other multisensory cortical areas to form a multisensory integrative network [56]. The majority of VM patients show an increase in the sensitivity to olfactory, auditory, and visual stimuli [57,58]. In this study, more than half of the VM patients were hypersensitive to somatosensory, visual, auditory, and olfactory stimuli during VM attacks. Thus, disrupted FC between the thalamus and these brain regions may contribute to multisensory integration abnormalities, which might disrupt the pathway used to process the multisensory, and induce hypersensitivity to external sensory stimuli in VM patients. We found that FC was not significantly correlated with clinical features in VM patients. This may be due to the inadequate accumulative effect of these clinical features on brain FC. If more patients with severe VM were included, the correlations of FC with the degree of vertigo and migraine may reach statistical significance.

## 5. Limitations

Although our work highlights the function of the thalamocortical region in VM patients, it has some limitations. First, although we used a strict diagnosis for VM that may have reduced participant variability and ensured the reliability of the results, the sample size was small. Second, we did not evaluate the brain functional alterations in subgroups of patients with migraine (without aura and with aura). Future studies should include subgroup analyses to elucidate the brain functional differences between the subgroups. Third, the majority of patients with VM in this study were female, and it may not have been excluded that sex differences may have influenced brain functional alterations in patients with VM. Future studies will be necessary to further identify how sex influences brain structure and FC.

## 6. Conclusions

This study revealed bilaterally abnormal whole-brain functional patterns in the thalamus and abnormal resting-state FC between the left thalamus and other cortical areas in VM patients. Furthermore, the decreased ALFF in the left thalamus negatively correlated with VM episode frequency. Our results indicated the important role of the thalamus in VM patients and suggested that the connectivity abnormalities of thalamocortical regions contributed to abnormal pain information processing and modulation, as well as transmission and multisensory integration in VM patients. 

## Figures and Tables

**Figure 1 brainsci-13-00183-f001:**
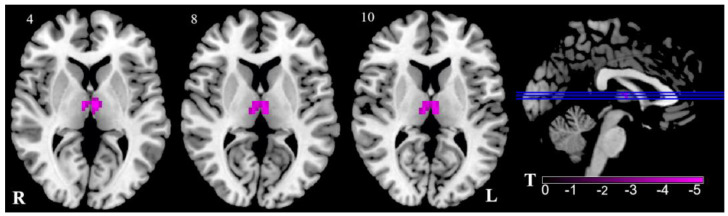
Regional ALFF reduction in VM patients in contrast to HC. Pink signifies decreased ALFF. The axial image was overlaid on the transverse section of the MNI-152 standard anatomical image. Blue line represents the layers in the sagittal. Numbers indicate z slice. The color scale denotes the t-value. R, right; L, left.

**Figure 2 brainsci-13-00183-f002:**
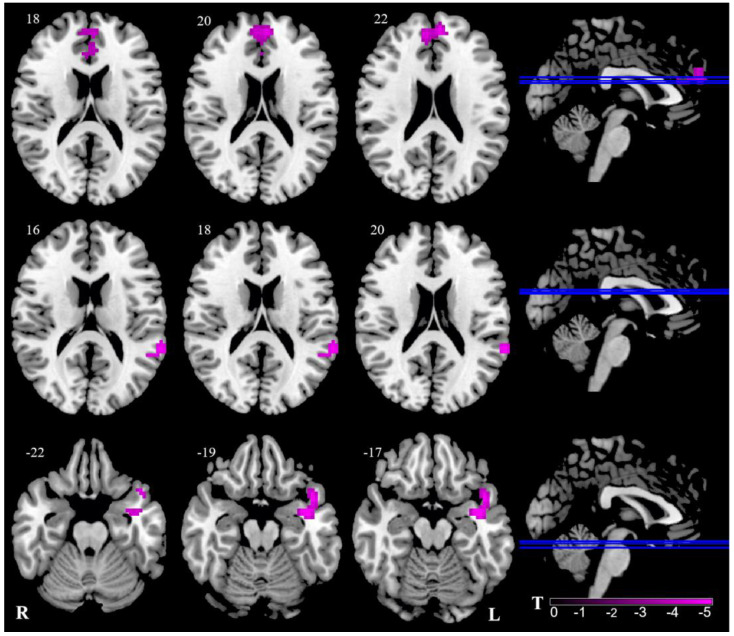
Abnormal FC of VM patients in contrast to HC. Pink signifies decreased ALFF. The axial image was overlaid on the transverse section of the MNI-152 standard anatomical image. Blue line represents the layers in the sagittal. Numbers indicate z slice. The color scale denotes the t-value. R, right; L, left.

**Figure 3 brainsci-13-00183-f003:**
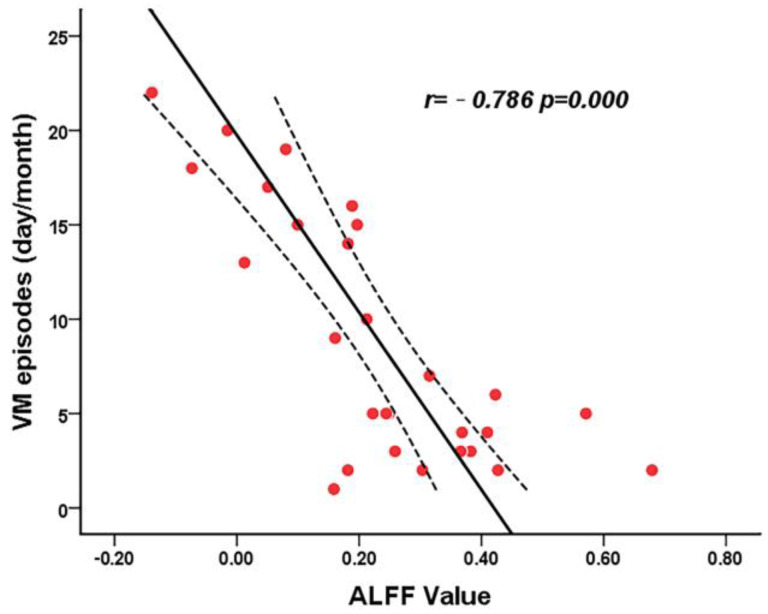
Correlation between the altered ALFF of the left thalamus and the frequency of VM attacks in patients with VM. Each Red dot denotes a sample. The solid line indicates the approximate line using partial correlation method. Smoothing curves represent the 95% of confidence intervals from the fit.

**Table 1 brainsci-13-00183-t001:** Demographic and clinical characteristics.

Characteristics	VM (*n* = 28) Mean ± SD	HC (*n* = 28) Mean ± SD	*p* Value
Sex (female/male) *	24/4	24/4	1.00
Age (years)	40.18 ± 10.26	38.25 ± 12.47	0.53
Education (years)	13.82 ± 3.65	15.00 ± 1.98	0.14
Disease duration (years)	8.68 ± 7.52		
VM Episode frequency (number)	8.82 ± 6.64		
Type of vertigo, *N (%)*			
Spontaneous vertigo	14/28 (50.00%)		
Positional vertigo	9/28 (32.14%)		
Visually induced vertigo	1/28 (3.57%)		
Head-motion-induced vertigo	4/28 (14.29%)		
Vomiting	10/28 (35.71%)		
Nausea	12/28 (42.85%)		
Photophobia	21/28 (75%)		
Phonophobia	19/28 (67.85%)		
VAS	5.07 ± 2.73		
MIDAS	47.46 ± 43.58		
HIT-6	51.86 ± 19.36		
DHI	47.71 ± 16.04		

* chi-square test.

**Table 2 brainsci-13-00183-t002:** Regions of reduced ALFF in VM patients compared with HC.

Brain Regions		Peak MNI			Cluster Voxels	*T*	*p*
		x	y	z			
R	Thalamus	6	−12	6	41	−4.83	0.000
L	Thalamus	−3	−12	6	43	−4.83	0.000

Note: MNI, Montreal Neurological Institute; R, right; L, Left.

**Table 3 brainsci-13-00183-t003:** Abnormal FC of left thalamus in VM patients compared with HC.

Seed Points	Brain Regions		BA	Peak MNI			Cluster Voxels	*T*
				x	y	z		
L Thalamus	L	Medial prefrontal cortex	9	−6	54	27	74	−5.36
	R	Medial prefrontal cortex	32	7	52	26	64	−5.36
	L	Anterior cingulum cortex	24	−2	36	18	10	−5.36
	R	Anterior cingulum cortex	32/24	5	48	22	15	−5.36
	L	Superior temporal gyrus	42	−63	−42	19	58	−4.25
	L	Middle temporal gyrus	21	−51	−48	3	66	−4.25
	L	Temporal pole	38	−42	−3	−18	58	−4.86

## Data Availability

The raw data that support the findings of this study are available on request from the corresponding author.

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
