# Peer review of "Decreased ALFF and Functional Connectivity of the Thalamus in Vestibular Migraine Patients"

_brainsci, 2023, doi:10.3390/brainsci13020183_

Round 1

Reviewer 1 Report

The authors of the publication "Decreased ALFF and Functional Connectivity of the Thalamus in Vestibular Migraine Patients" conduct a research study to examine Vestibular migraine (VM) utilizing a functional connectivity measure and statistical analysis contrasting healthy individuals with VM patients.

This study sounds intriguing to me, and the research is broadly accessible.

Comments

·      There is a typo on line 9.

·      Introduction: I suggest providing a brief overview of the work done for this research.

·     Introduction: A few sentences indicating the structure of the document might be included for information.

·   Where the authors say, "To allow the patients to adjust to the magnetic field," on lines 132–133, is unclear.

·      "Accompany symptoms" in line 192 doesn't seem appropriate.

·  The tab captions might need more information; they're not very informative. For the sake of paper, the text's explanation of the tabs may also be increased. I also suggest discussing on the meaning of the Table labels (such as the Peak MNI).

·  The word "ear irrigation" sounds awful in line 239.

·The following statement by the authors, "The results of our present study show that the parts of thalamocortical pathways with decreased FC might disrupt and interfere with information flow between the thalamus and cortex with consequent disturbances in multisensory information integration, pain modulation, and processing in patients with VM," is not clear how they intend to be interpreted. Furthermore, it is unclear how thalamocortical circuits may be examined using the methodology described in this work.

·  The authors' hypotheses for lines 282-285 and 295-297 are not quite obvious.

·    Lines 308–310 are unclear; I suggest refocusing.

·   Testing the same subject group utilizing TMS-EEG in an experimental context to track connection changes might be an intriguing future research (TMS ref: Real-time artifacts reduction during TMS-EEG co-registration: a comprehensive review on technologies and procedures)

Author Response

The authors of the publication "Decreased ALFF and Functional Connectivity of the Thalamus in Vestibular Migraine Patients" conduct a research study to examine Vestibular migraine (VM) utilizing a functional connectivity measure and statistical analysis contrasting healthy individuals with VM patients.

This study sounds intriguing to me, and the research is broadly accessible.

 Response: We are also grateful to you for the instructive and constructive as well as positive comments and suggestions. Accordingly, we have revised the manuscript earnestly and cautiously. All amendments have been highlighted in red in the revised manuscript. In addition, point-by-point responses to the comments are listed below in this letter.

Comments

  1. There is a typo on line 9.

 Response: Thank you for your precious comments. Correction has been made in the revised manuscript.

  1. Introduction: I suggest providing a brief overview of the work done for this research.

Response: Thanks for raising this critical issue. With the advice from your suggestion, corrections have been made in the revised manuscript (Lines 85-90)

  1. Introduction: A few sentences indicating the structure of the document might be included for information.

Response: Thanks for raising this important issue. With the advice from your suggestion, corrections have been made in the revised manuscript (Lines 85-90)

  1. Where the authors say, "To allow the patients to adjust to the magnetic field," on lines 132–133, is unclear.

Response: Thank you very much for your constructive comments. Correction has been made in the revised manuscript.

  1.  "Accompany symptoms" in line 192 doesn't seem appropriate.

Response: Thank you for your appropriate comments. Correction has been made in the revised manuscript.

  1. The tab captions might need more information; they're not very informative. For the sake of paper, the text's explanation of the tabs may also be increased. I also suggest discussing on the meaning of the Table labels (such as the Peak MNI).

Response: Thanks for your thoughtful suggestion. Correction has been made in the Table 2 and 3 in the revised manuscript.

  1. The word "ear irrigation" sounds awful in line 239.

Response: Thanks for reminding me of this issue. With the advice from your suggestion, corrections have been made in the revised manuscript.

  1. The following statement by the authors, "The results of our present study show that the parts of thalamocortical pathways with decreased FC might disrupt and interfere with information flow between the thalamus and cortex with consequent disturbances in multisensory information integration, pain modulation, and processing in patients with VM," is not clear how they intend to be interpreted. Furthermore, it is unclear how thalamocortical circuits may be examined using the methodology described in this work.

Response: Thanks for raising this critical issue. With the advice from your suggestion, corrections have been made in the revised manuscript (Lines 318-319)

  1.  The authors' hypotheses for lines 282-285 and 295-297 are not quite obvious.

Response: Thanks for raising this critical issue. With the advice from your suggestion, corrections have been made in the revised manuscript (Lines 284-286ï¼›Lines 318-310)

  1.  Lines 308–310 are unclear; I suggest refocusing.

Response: Thanks for raising this critical issue. With the advice from your suggestion, corrections have been made in the revised manuscript (Lines 85-90)

  1.  Testing the same subject group utilizing TMS-EEG in an experimental context to track connection changes might be an intriguing future research (TMS ref: Real-time artifacts reduction during TMS-EEG co-registration: a comprehensive review on technologies and procedures)

Response: Thank you for your positive comments on the present study and insightful suggestions on further investigation. Future studies should utilize the TMS-EEG to study patients with vestibular migraine.

Reviewer 2 Report

Dear Authors, 

In this manuscript, the authors investigated the differences in the amplitude of low-frequency fluctuation (ALFF) and analyzed whole-brain functional connectivity (FC) in vestibular migraine (VM) patients. Moreover, they examined the relationship between clusters and clinical features by correlation analysis.

Based on their results, the connectivity abnormalities of the thalamocortical region contribute to abnormal pain information processing and modulation, transmission, and multisensory integration in patients with VM.

The topic is timely and may attract much attention.

I have only a few suggestions:

1. Space errors occur in several places. Please, review the text carefully and correct these errors. 

e.g.: Line 74 ( 2014)

2. For references in parentheses, there is no space after the semicolon, which is unpleasant when reading.

3. I recommend the authors use more references to support their claims. I believe that adding more citations will help to provide better and more accurate background to this study. 

4. In the Materials and Methods part, the number of participants in the experiment is not clear. Probably just a typo, but please correct it.

Line 93 "A total of 24 right-handed VM patients (4 with aura and 24 without aura)"

Also, I have a few questions:

1. "Six VM patients received medications for preventing migraine as well as nonsteroidal analgesics." - These drugs were also given during the study or were used by the patients only during the attacks? Did they see any difference in these patients compared to the others?

2. If I understand correctly, there was no difference between men and women in terms of clinical symptoms. At the same time, the number of men included in the experiment is very low compared to the number of women, so in my opinion it is not possible to make a fair comparison in this way. Do they have any previous data on the gender difference obtained by their own or other research groups? Are such comparisons planned in the future?

Author Response

In this manuscript, the authors investigated the differences in the amplitude of low-frequency fluctuation (ALFF) and analyzed whole-brain functional connectivity (FC) in vestibular migraine (VM) patients. Moreover, they examined the relationship between clusters and clinical features by correlation analysis.

Based on their results, the connectivity abnormalities of the thalamocortical region contribute to abnormal pain information processing and modulation, transmission, and multisensory integration in patients with VM.

Response: We would like to express our sincere gratitude to your concern over our paper, and to you for your constructive and positive comments and suggestions.

The topic is timely and may attract much attention.

 I have only a few suggestions:

  1. Space errors occur in several places. Please, review the text carefully and correct these errors. 

e.g.: Line 74 ( 2014)

Response: Correction has been made in the revised manuscript.

  1. For references in parentheses, there is no space after the semicolon, which is unpleasant when reading.

Response: All corrections concerned have been made in the revised manuscript.

  1. I recommend the authors use more references to support their claims. I believe that adding more citations will help to provide better and more accurate background to this study. 

Response: Thanks for your thoughtful suggestion. Some statements and citations

 have been added in the introduction of the revised manuscript (Lines 81-88)

  1. In the Materials and Methods part, the number of participants in the experiment is not clear. Probably just a typo, but please correct it.

Line 93 "A total of 24 right-handed VM patients (4 with aura and 24 without aura)"

 Response: Thanks for your thoughtful suggestion. Correction has been made in the revised manuscript.

Also, I have a few questions:

  1. "Six VM patients received medications for preventing migraine as well as nonsteroidal analgesics." - These drugs were also given during the study or were used by the patients only during the attacks? Did they see any difference in these patients compared to the others?

 Response: Thanks for raising this critical issue raised in the feedback. These drugs were used by the patients only during the attacks in the past. To avoid any possible pharmacological interference with BOLD signal changes, patients with VM did not take medications for at least 3 days before the fMRI scan. Accordingly, a statement has been added in the materials and methods of the revised manuscript. There was no difference between patients with medication and patients without medication in terms of clinical symptoms. Due to the fact that patients who   received the medication are few, there was no means to carry out brain functional study in this study.

  1. If I understand correctly, there was no difference between men and women in terms of clinical symptoms. At the same time, the number of men included in the experiment is very low compared to the number of women, so in my opinion it is not possible to make a fair comparison in this way. Do they have any previous data on the gender difference obtained by their own or other research groups? Are such comparisons planned in the future?

 Response: Thank you for your precious comments. Due to the fact that the majority of patients with VM were female, and sex differences may have influenced GM volume and FC alterations in patients with VM. Future studies will be necessary to further identify how sex influences brain structure and FC. This limitation has been added in the limitation section of the revised manuscript.

Reviewer 3 Report

The authors compared ALFF between VM patients and their age and gender matched healthy controls. The significant ALFF clusters were further used as the seed regions for resting-state functional connectivity (rsFC). The rsFC was then compared between the groups. The associations between clinical data and those ALFF and rsFC that showed patient-control difference were explored in VM patients. They found ALFF at bilateral thalamus was decreased in VM patients compared with healthy controls, and rsFC between left ALFF-seeded and bilateral mPFC, bilateral ACC, left superior/middle temporal gyrus, and the temporal pole were weaker in VM patients. The ALFF at left thalamus was also associated with the frequency of VM attacks in VM patients.

This is a well-written manuscript with sound study design and moderate sample sizes in both groups. The statistical methods are appropriate. The topic is of clinical importance and it may add critical knowledge to the field. I have a few minor comments as follows:

Page 3, line 136. The exclusion criterion of <0.2 mm volume-level mean framewise displacement might be ‘>0.22mm’.

For 3.2. ALFF results section, are bilateral thalamus the only clusters significantly showing ALFF differences between the two groups? If so, please state this in this section.

For 3.3 Seed-based FC results, are there any right thalamus seeded rsFC that differed between the two groups? If so, please add this to the results.

For Figure 3, it is recommended switching the x axis and y axis, i.e., ALFF value as X and WM episodes as Y.

Author Response

The authors compared ALFF between VM patients and their age and gender matched healthy controls. The significant ALFF clusters were further used as the seed regions for resting-state functional connectivity (rsFC). The rsFC was then compared between the groups. The associations between clinical data and those ALFF and rsFC that showed patient-control difference were explored in VM patients. They found ALFF at bilateral thalamus was decreased in VM patients compared with healthy controls, and rsFC between left ALFF-seeded and bilateral mPFC, bilateral ACC, left superior/middle temporal gyrus, and the temporal pole were weaker in VM patients. The ALFF at left thalamus was also associated with the frequency of VM attacks in VM patients.

This is a well-written manuscript with sound study design and moderate sample sizes in both groups. The statistical methods are appropriate. The topic is of clinical importance and it may add critical knowledge to the field. I have a few minor comments as follows:

Response: Thank you for your positive comments on the present study and insightful suggestions.

  1. Page 3, line 136. The exclusion criterion of <0.2 mm volume-level mean framewise displacement might be ‘>0.22mm’.

  Response: Thank you for your timely comments. Correction has been made in the revised manuscript.

  1. For 3.2. ALFF results section, are bilateral thalamus the only clusters significantly showing ALFF differences between the two groups? If so, please state this in this section.

 Response: Thanks for your valuable suggestion. The bilateral thalamus are the only clusters significantly showing ALFF differences between the two groups. With the advice from your suggestion, corrections have been made in the revised manuscript.

  1. For 3.3 Seed-based FC results, are there any right thalamus seeded rsFC that differed between the two groups? If so, please add this to the results.

Response: Thanks for raising this critical issue. we found no significant FC alterations in the right thalamus and the whole brain.

  1. For Figure 3, it is recommended switching the x axis and y axis, i.e., ALFF value as X and WM episodes as Y.

Response: Thank you very much for your constructive comments. According to your suggesting, the x axis and y axis have been switched for in Figure 3 in the revised manuscript.

Round 2

Reviewer 1 Report

The paper has been improved by the authors' revisions. They handled all of my earlier concerns, thus I don't have any other feedback to offer.